# Welfare Issues on Israeli Dairy Farms: Attitudes and Awareness of Farm Workers and Veterinary Practitioners

**DOI:** 10.3390/ani11020294

**Published:** 2021-01-24

**Authors:** Sarah Weyl-Feinstein, Yaniv Lavon, Noa Yaffa Kan, Meytal Weiss-Bakal, Ayelet Shmueli, Dganit Ben-Dov, Hillel Malka, Gilad Faktor, Hen Honig

**Affiliations:** 1Veterinary Services, Ministry of Agriculture and Rural Development, P.O.B. 12, Bet Dagan, Hamakabim St., Rishon Letzion 7519701, Israel; MeytalW@moag.gov.il (M.W.-B.); Ayelets@moag.gov.il (A.S.); dganitb@moag.gov.il (D.B.-D.); 2Israel Cattle Breeders’ Association, P.O.B. 3015, Caesarea Industrial Park 38900, Israel; yaniv@icba.co.il; 3Mitrani Department of Desert Ecology, The Jacob Blaustein Institutes for Desert Research, Ben Gurion University of the Negev, P.O.B. 37, Midreshet Ben Gurion 84990, Israel; noayaffakan@gmail.com; 4Extension Service, Ministry of Agriculture and Rural Development, P.O.B. 30, Bet Dagan, Hamakabim St., Rishon Letzion 7519701, Israel; hilmal@shaham.moag.gov.il; 5Hachaklait Veterinary Services Ltd. Corporation, Bareket St. 20, Caesarea 3097020, Israel; faktorvet@gmail.com

**Keywords:** behavior, dairy cows, farm workers, veterinary practitioners, pain, calves

## Abstract

**Simple Summary:**

Animal welfare science embraces all factors that might affect the physical and emotional state of the animal, its ability to cope, and its overall quality of life. In recent years, awareness of farm animal welfare has increased among veterinary practitioners—a major professional figure influencing a farm’s routine, farm workers, consumers, and the general public. In particular, the farm worker’s knowledge of animal welfare is an essential component of the rearing system. The aim of this study was to examine attitudes toward and awareness of select animal welfare issues among farm workers and practitioners. A survey was performed based on anonymous questionnaires filled out by dairy farm workers and veterinary practitioners. The results demonstrated that farm workers’ enjoyment of their work is of great importance, as is their cows’ welfare. The survey showed the farm workers’ awareness of their influence on the cow during milking, the effects of stress on milk production, and the possible effect of human behavior on heifers and cows. The main areas where animal welfare might be improved were farmers’ awareness of learning, memory, and pain masking in cattle, and knowledge transfer from veterinary practitioners to the farm workers. The survey answers further emphasized the crucial importance of communication and understanding between farm workers and their practitioners.

**Abstract:**

Attitudes toward practical dairy cow welfare issues were evaluated based on a questionnaire answered by 500 dairy farm workers and 27 veterinary practitioners. Primarily, the effect of demographic characteristics on attitudes toward cattle welfare was tested. Professionally, five themes were identified: effect of welfare awareness on productivity, knowledge of cattle’s senses and social structure, effects of man–animal interactions on milk yield, pain perception and prevention, and knowledge transfer from veterinary practitioners to farm workers. Farms with a higher welfare awareness score also had higher annual milk yield, with an annual mean difference of 1000 L of milk per cow between farms with higher and lower awareness scores. Veterinary practitioners showed high awareness of cows’ social structure, senses, and pain perception. Farm workers were aware of the influence of man–animal interactions during milking and stress effects on milk yield, and the possible effect of man’s behavior on heifers and cows. Practitioners and farm workers had different views regarding pain perception, mostly involving mutilation procedures. All veterinary practitioners advocated the use of pain alleviation in painful procedures, but only some of them instructed the farm workers to administer it. The survey results emphasize the variation in welfare knowledge and practical applications across farms, and the interest of both the animals and their managers to improve applied knowledge of best practice.

## 1. Introduction

The Israeli dairy sector consists of about 135,000 Israeli-Holstein cows on 700 dairy farms. The dairy sector is well organized by the Israeli Milk Board and the Israeli Cattle Breeders’ Association. Based on the Israeli Herd Book data, which includes nearly 90% of the dairy cattle, the 2020 average annual milk production was 12,120 kg/cow, with 3.73% fat and 3.25% protein [1,2]. The Israeli dairy sector is divided into two main subsectors: cooperatives, which have relatively large dairy herds (mean 450 cows per unit; 164 farms), representing 62% of the cows with recorded production, and which predominantly participate in milk recording; and relatively small family dairy farms (mean 114 cows per unit; 536 farms), approximately 75% of which participate in milk recording, and representing 38% of the cows with recorded production [2,3]. In Israel, the veterinary practitioner is the main professional on the farm, visiting family farms once to twice a week, and cooperative farms twice to three times a week. Most of the farm workers’ actions are decided upon with the practitioner, including management and ethical decisions. The dairy farms are managed by an Israeli farm worker, and the other workers are both local and foreigners, mainly from Thailand. Farm workers can attend a variety of voluntary training programs from governmental organizations and private companies [2].

Animal welfare remains an area of consistent public concern, with acceptance of animal sentience enshrined in the legislation of many countries [4]. A draft of the Israel Animal Welfare Regulations (Protection of Animals; Holding Calves) has been recently released for public comments. When the authors started writing the regulations in 2014, limited data were available on the extent of welfare knowledge, social rearing preferences, mutilation practices, and welfare attitudes and perceptions in the Israeli dairy sector. The Veterinary Services felt that a national database of concurrent knowledge would be the foundation for successful implementation of the regulations. 

Many data are collected through precision dairy farming [5,6]. However, the data generated by these systems pertaining to the health and welfare of livestock are still mostly unused, with dairy farmers, employees, and advisors requiring more training [7]. New sensors and management software are already available on most farms in Israel, providing a large amount of online data daily. The veterinary practitioner and farm manager should be adequately trained to make the right decisions based on these data [8]. However, in Israel, little is known about veterinary practitioners’ knowledge of, or approach to, cows’ senses, behavior, learning, and memory. 

Understanding the differences between human and bovine senses can contribute to the way in which the cattle are treated. For example, cows hear a wider range of frequencies than humans do (20,000–40,000 Hz) [9]. Dairy farmers’ awareness of this could improve man–animal interactions, resulting in calmer handling and milking. Understanding cattle’s social structure could be beneficial to proper farm routine [10,11]. Previous studies have shown that social acclimation following group mixing has an important impact on cattle, with calves’ behavioral acclimation taking up to 3 weeks [12], and resumption of dairy cows’ daily milk production to pre-mixing levels taking up to 6 weeks [13]. Understanding the importance of cattle’s social structure can decrease stress and improve welfare in the daily routine.

Research into animals’ mental status, i.e., their sensory and cognitive abilities, has developed greatly in recent years [14]. Cognitive mechanisms, which include sensory ability, learning, memory, and decision making [15], allow the animal to cope with the environment in flexible ways. As a sentient being, a cow’s awareness can be tested by its ability to assess a particular event concerning itself [16]. The combination of sensing and awareness is at the root of pain processes, data processing, and emotions [17]. Thus, mental, psychological, and cognitive needs play an essential role in the well-being of the dairy cow [18]. By testing motivation and decision making, an animal’s emotional state can be determined, as expressed in behavioral and physiological changes [19,20]. Therefore, these changes can serve as animal-based welfare indicators. Productivity is one of the commonly used indicators for welfare assessment [21].

Dairy farm workers’ behavior seems to affects cows’ behavior and milk yield [22]. Cows can differentiate between farm workers according to their behavior [23] and farm workers’ behavior has been shown to affect milk production [24]. Many stress-causing factors can influence animal behavior, which in turn affects the neuro-hormonal reflex, leading to inhibition of milk ejection and production [25], as well as increases in milk somatic cell count and bacterial count (CFU) [26]. Proper guidance of farm workers regarding behavior while milking cows has been shown to increase milk yield and protein and fat levels compared to levels achieved with untrained workers [27]. In addition, the ability to distinguish between dairy farm workers based on their behavior has also been documented in calves [28].

Findings demonstrate that learning at an early age can impact cows’ behavior as adults [29]. For example, social husbandry of young animals has effects on their response to stress [30], infection courses [31], and wound healing [32]. Social interactions at a young age resulted in better coping with changes throughout life, making them easier to manage [33].

Freedom from pain is one of the main aspects of animal welfare [34]. Among veterinary practitioners, there is general agreement that farm animals can suffer from painful conditions during the daily farm routine [35,36], and pain alleviation can be improved on farms [37]. Studies have shown that the use of pain relief depends directly on the ability of professionals to correctly assess pain, rather than on economic or other considerations for the administration of analgesics [38,39]. Thus, the first step toward improvement is the recognition and awareness of the farmers to these painful situations.

In light of the above, we performed a survey of dairy farm veterinary practitioners and workers to examine current attitudes to, and awareness of, cattle’s mental needs, and to identify the areas where further awareness of welfare issues can be improved.

## 2. Methods

### 2.1. The Veterinary Practitioner Survey

A total of 41 Israeli dairy cow veterinary practitioners working with the corporation Hachaklait Veterinary Services Ltd. received an anonymous questionnaire through internal software in 2018. The questionnaire, which is provided in Appendix A, contained 6 personal questions and 20 closed questions on different welfare issues. The questionnaire was written by veterinarians working for the Veterinary Services, and the Extension Service of the Ministry of Agriculture, and was further edited by professionals from the Israeli Milk Board and Cattle Breeders’ Association. In the first part of the survey, the practitioners were asked for their age, gender, work seniority, and training background on welfare issues. They were then asked to score the extent of their enjoyment from their work on a scale of 1 (low enjoyment) to 5 (very high enjoyment), and the importance of animal welfare in their practice from 1 (low) to 5 (very high). In the second part of the questionnaire, they were asked to choose the most accurate answer to questions relating to five themes: effect of welfare awareness on productivity, knowledge regarding cattle’s senses and social structure, effect of man–animal interactions on milk yield, pain perception and prevention, and knowledge transfer from the veterinary practitioner to the farm worker.

### 2.2. The Farm Worker Survey

A questionnaire was written by the authors, translated into Arabic, English and Thai, and distributed by hand to dairy farm workers on each farm. The questionnaire is provided in Appendix A. The answer format consisted predominantly of closed questions, while some questions provided the opportunity for further free text comments. All workers were present at the farm on the scheduled day with the survey conductor, and filled out the questionnaire voluntarily and anonymously (only giving the farm’s name). Only 3% of the farm workers asked to answer the survey refused to participate. Since some of the farm workers do not use electronic communication, hand-filled questionnaires ensured proper representation of all employees, instead of only the Hebrew speakers and managers. 

Dairy farms were categorized as cooperatives or private family farms (as mentioned in the introduction). In total, 61 cooperative and 113 family dairy farms were selected from all geographic regions, representing 37% and 20% of each category, respectively, in the national herd book, which consists of all the Israeli dairy farms’ database. The first part of the survey asked general demographic questions: responder’s gender, age, seniority in farming (length of time working on a dairy farm), academic background, past guidance on welfare topics, role on the farm, extent of enjoyment from the work, and extent of the importance of animal welfare. 

The second part asked questions related to five themes: effect of welfare awareness on productivity, knowledge regarding cattle’s senses and social structure, effect of man–animal interactions on milk yield, pain perception and prevention, and knowledge transfer from veterinary practitioners to farm workers. Answers were single choice only. Adaptation to each sector resulted in slight differences between the questions for the veterinary practitioners and those for the farm workers. Specific questions were chosen based on an assessment of the concurrently existing knowledge and suspected gaps. Previous audits and guidance by the extension unit of the Ministry of Agriculture in recent years revealed the subjects where knowledge was suspected to be lacking.

To facilitate the ranking of extent of enjoyment from work and personal importance of welfare issues, we used a 1–5 scale (1—very low, 5—very high). To evaluate the farm workers’ perception of animals’ pain, respondents were asked to score dairy cows’ expression of pain on a scale of 1 (do not hide) to 5 (hide excessively).

Data from the completed questionnaires were collected and analyzed; 18 questions received “correct” and “incorrect” answers, and questions 5 and 13 were excluded. “Correct” answers were given 5.555 points, whereas “incorrect” was given zero. 

An overall welfare assessment score (WAS) was given to each respondent to examine the effect of attitude and awareness on different parameters. Additional analysis was performed on farms reporting to the Herd Book (*n* = 166): a mean WAS of the workers on each farm was calculated, so every farm received a final farm welfare assessment score, designated FWAS. The effects of these FWAS were analyzed with the farms’ production data (taken from the Israeli Herd Book).

### 2.3. Statistical Analysis

Responses were entered in standard spreadsheet software (Excel 2016, Microsoft corporation, Redmond, Washington, USA). Statistical analysis of the data was performed using SAS software (version 9.2, SAS Institute, Cary, NC, USA). All variables’ means were calculated using the PROC MEANS procedure of SAS. In addition, we used a general linear mixed model (GLM) and the PROC MIXED procedure of SAS to test the effects of different parameters on the participants’ answers (both veterinary practitioners and farm workers) and their WAS. The entire model for the participants’ final grade was: Herd + welfare guidance + duty on the dairy farm + seniority + error, where Herd = 166 dairy farms; welfare guidance = yes/not at all; duty on the dairy farm = milking personnel, responsible for nursery, responsible for health, and general manager; seniority ≤ 4 years, 4–14 years, 15–29 years, and >30 years. All variables except Herd were considered fixed effects. Herd was included as a random effect.

A second analysis was performed to test the effect of FWAS on cow milk production over 305 days. Only the 166 farms reporting to the Herd Book were included. We separated the dairy farms into four groups according to their FWAS values: group 1, ≤50; group 2, 50.1–67; group 3, 67.1–89; group 4, 89.1–100. The entire model was: 305 d milk yield, 305 d ECM, Fat %, Protein %, SCC = Herd + dairy farm size + FWAS group + error, where Herd = 166 farms reporting to the Herd Book; dairy farm size ≤ 300 or >300 cows; and FWAS group = four groups based on FWAS as delineated above. The results are presented as least squares (LS) mean values ± SE. To compare levels within a variable, we ran the Bonferroni adjustment for multiple comparisons.

## 3. Results

### 3.1. General Characteristics

#### 3.1.1. Veterinary Practitioners

A total of 27 veterinary practitioners, representing 66% of the bovine practitioners working at Hachaklait Veterinary Services, answered the questionnaire (24 men and 3 women). Hachaklait Veterinary Services works with nearly 90% of the dairy farms in Israel. Of the surveyed practitioners, 20 declared that they had undergone some animal welfare seminars or lectures, while the other 7 had little exposure to the topic. No correlation was found between the practitioner’s gender and the overall scoring, or between age, seniority, or training experience and their personal score. The extent of the practitioners’ enjoyment from their work is shown in Figure 1, and the results of the extent of importance of animal welfare to them are shown in Figure 2. The distribution of the practitioners’ answers regarding senses, memory, and pain perception are presented in Table 1.

#### 3.1.2. Farm Workers

A total of 500 Israeli farm workers from 174 dairy farms participated in the survey: 324 farm workers were from cooperative dairy farms, and 176 were from private family farms; 443 were men, 57 women; 79 (16%) of the responders were workers from Thailand. In the cooperative farms, the most frequent category (39%) was the young workers (aged < 30 years; *n* = 124), whereas in the family farms, the most frequent (39%) was the oldest category (57+ years; *n* = 66). A significant part of the workers (35%; *n* = 113) on the cooperative farms have less than 4 years of work experience at the farm, while at the family farms, 44% work more than 30 years in farming (*n* = 75).

The first question referred to the extent of personal enjoyment from work at the farm, and the results are presented in Figure 1. Farm workers gave high importance to welfare issues, similar to the practitioners (Figure 2). The farm workers’ answers regarding enjoyment from work and importance of animal welfare were positively correlated (*r = 0.56*), i.e., those who enjoyed their work were also concerned about animal welfare.

### 3.2. Effects of Demographic Variables on Farm Workers’ Attitude toward and Awareness of Cattle Welfare

The younger workers (aged < 30 years; *n* = 129) reached a mean WAS of 62.7 ± 3.2, similar to the workers aged 31–41 years (*n* = 92) who had a mean WAS of 62.5 ± 3.3. The workers aged 42–56 years (*n* = 113) had the highest WAS, 66 ± 3, and the older workers (57+ years; *n* = 95) scored slightly lower, with an average WAS of 60 ± 3.3, (*p* = 0.09).

Farm workers on private family farms (*n* = 155) tended to score slightly lower than those working on cooperative dairy farms (*n* = 274), with mean WAS of 61.2 ± 2.9 and 64.4 ± 2.8, respectively (*p* = 0.08). Farm workers’ formal education significantly affected the survey score: farm workers with no academic or technical education (*n* = 224) scored 60.3 ± 2.7, those who graduated from professional or technical studies (*n* = 122) scored 61.6 ± 2.9, and academic graduates (*n* = 83) had a WAS of 66.5 ± 3.2 (*p* = 0.01).

On-farm self-training for the work with dairy cows led to lower WAS than training through professional husbandry courses and workshops. Farm workers who were trained locally on the farm (*n* = 222) had a mean WAS of 59.3 ± 3, whereas farm workers who attended external courses (*n* = 207) scored 66.4 ± 2.7 (*p <* 0.0001). In addition, the extent of guidance on “cattle welfare” significantly affected workers’ knowledge, as expressed by their WAS (Figure 3). Further analysis of the differences between the two sectors showed that 55% of the workers in the cooperative farms were guided in subjects regarding cattle’s welfare and behavior, whereas only 33% of the workers from the family farms received such guidance. The sources of guidance attended by farm workers were divided into 33% professional government courses and lectures, 21% private organizational guidance for practical welfare improvement, 7% Hachaklait Veterinary Services guidance, and 39% from other sources.

Personal WAS on the survey differed among those with different farming duties. The farm worker responsible for the health of the herd had the highest score, followed by the general manager, the worker responsible for the nursery (young calves), and finally, the milkers, who had the lowest WAS (Figure 4).

Seniority of less than 4 years resulted in relatively low WAS. Numerous years of experience had a positive effect on the level of farm workers’ knowledge regarding animal welfare issues: from 4–14 years and 15–29 years of experience, up to 30 years of seniority. However, after 30 years, the mean WAS decreased (*p <* 0.0001; Figure 5).

### 3.3. Links between Farm Workers’ Awareness of Welfare and Milk Yield 

The higher-producing dairy farms had a higher mean FWAS (Figure 6). Farms that scored under 50 on the survey had a mean annual milk yield of 11,167 ± 250 L/cow, whereas farms scoring 89–100 had a mean yearly milk yield of 12,162 ± 259 L/cow. No effect of FWAS was seen on milk fat or protein content, or on somatic cell count.

### 3.4. Farm Worker Awareness of Cattle Social Structure and Effect of Separation on Conception Rates

Farm workers and veterinary practitioners were asked to estimate the duration of social acclimation for a cow introduced into a new group of unfamiliar cows. Both gave similar answers, corresponding to previous findings (Figure 7).

Another question on the survey, directed to the farm workers, referred to the social hierarchy in cattle: “Will the cow’s level of dominance affect its position in the herd?”; 77.6% of the workers answered “yes” (*n* = 378), 8.4% (*n* = 41) did not know, 8.2% (*n* = 40) answered that “there is no hierarchy in cow herds,” and 5.7% (*n* = 40) answered “no.” Both parties were asked about the possible effect of separating a young cow for artificial insemination on its conception rate; 51% (*n* = 238) of the farm workers answered “it should not have any effect,” 37.4% (*n* = 175) thought it might have a negative effect, and 11% (*n* = 52) assumed that it might have a positive effect. However, 63% (*n* = 17) of the practitioners thought that inseminating a cow that is restrained away from the others would negatively affect conception rate, 37% (*n* = 10) said that it would not have any effect, and no practitioner thought that it might have a positive effect on conception.

### 3.5. Farm Workers’ Awareness of Cattle’s Senses

Farm workers’ answers to the questions referring to cows’ sight and hearing are reported in Table 2.

### 3.6. Attitude toward and Awareness of Learning and Memory in Cattle and Man–Animal Interactions

The results of the questions exploring farm workers’ and veterinary practitioners’ perceptions of cows’ learning processes and memory, and of man–animal interactions, are given in Table 1 (practitioners’ answers) and Table 2 (farm workers’ answers).

### 3.7. Farm Workers’ Perceptions of Welfare and Milk Yield

Farm workers’ answers to questions exploring the effect of man–animal interactions on milk production are listed in Table 2. Furthermore, 86% (*n* = 405) of the farm workers thought that the way in which they lead the cows to the milking parlor can affect milk yield, whereas 14% (*n* = 66) thought that this has no effect at all. The results showed that 74.5% (*n* = 353) of the farmers administer medications to the cows in a designated shed (as instructed), while 17% (*n* = 65) do not use a specific location; 12% (*n* = 56) of the workers treated the cows in the milking parlor, which goes against professional recommendations. The question of whether high milk production signifies good welfare is elaborated on in Table 3.

### 3.8. Assumptions Regarding Pain Perception

Veterinary practitioners’ and farm workers’ results are given in Table 1 and Table 3, respectively, demonstrating some small differences (that even if not statistically different are worth mentioning). The perception of pain expression in cows is demonstrated in Figure 8.

Regarding the need for pain relief following mutilation procedures, all practitioners and 71% of the farm workers thought that disbudding (using caustic paste) should be performed with pain relief medication. Only 63% of the farm workers thought that cold branding is a painful procedure, whereas 92.6% of the practitioners believed that this procedure requires pain medication.

### 3.9. Calves’ Social Structure

The use of different social rearing methods for calves was distributed as follows: 19.3% (*n* = 91) of farms moved their calves into groups at the age of 1 week, 22.7% (*n* = 107) at 2 weeks, 20% (*n* = 92) at 1 month, and 38.5% (*n* = 182) at the age of 2 months (at weaning). When asked about the ideal age for group rearing (Figure 9), 88% of the practitioners believed that calves should be reared in pairs or groups up to weaning, as opposed to only 48.5% of the farm workers. Interestingly, 41% of the farm workers thought that keeping pre-weaned calves in individual cages is preferable.

### 3.10. Knowledge Transfer

The vast majority (96%) of veterinary practitioners stated that they had instructed farm workers to apply medical therapy in a designated shed, rather than in the milking parlor, to prevent a negative association between pain and milking. However, only 74.5% of the farm workers followed those instructions. Moreover, 92.6% of the practitioners thought that cold branding requires pain medication, but only 63% of them instructed the farm workers to use it (Table 1).

The practitioners were asked what they do when a repeated welfare problem at the farm remains unsolved; 28% answered that they only make a remark to the farm worker, 34% said that they insist on solving the problem without further professional assistance, and 38% said that they turn to other professionals for a solution.

## 4. Discussion

The survey results demonstrated the importance of guidance and knowledge acquisition on farm workers’ perception of animal welfare issues. In a survey performed in the UK, 56% of the farmers stated that they had gained a major part of their farming skills from their family, and only 18% had learned their skills from agricultural courses [40]. In this study, 51% of the total farm workers had participated in external husbandry courses and 46% of them had received specific welfare and behavior guidance; these farm workers had enhanced welfare knowledge, as expressed by their WAS. However, only 33% of the family farm’s workers received guidance on welfare topics, as opposed to 55% of the workers from the cooperative farms. Seniority had a positive impact on knowledge, but only up to 30 years of experience. There are several possible explanations for these differences: according to survey results, the structure of family farms (accounting for 65% of the farms in the survey) consists of 39% senior owners (57+ years), who also serve as managers and usually work alone or with one additional worker, resulting in a limited workforce. In addition to the dairy farm, the owner is involved in other occupations and therefore has limited spare time for learning. In contrast, farm workers on cooperative farms are numerous and their duties are well separated among farm sections (nursery, herd health, nutritionist, etc.), and therefore, expertise on those farms is warranted. Furthermore, economic management of the family farms differs greatly from that of the cooperative ones. All of these factors make it difficult for the family farm workers to leave their farms for outside training. Hence, knowledge remains at a relatively basic level and the farm worker lacks up-to-date information. The lack of relevant knowledge may explain the tendency toward a scoring gap between cooperative and private family farms. A creative solution to this problem might be the farmer field school model, a concept for farmers’ learning, knowledge exchange, and empowerment that has been successfully applied in developing countries [41]. A survey examining how to enhance communication between farmers and veterinarians in the Netherlands showed that the most important aspects of such communication are a proactive approach, personalization of messages, providing a realistic frame of reference for the farmer, and use of the farmer’s social environment. They further stressed that social pressure through the presentation of successful examples from other farms may help to activate “hard-to-reach” farmers. [42].

Most dairy farmers, animal and dairy scientists, and many veterinarians argue that high levels of milk production and good health are clear evidence of high standards of welfare [43]. Enhanced knowledge in this domain generally contributes to a farm’s success, and our results specifically bolstered this argument: farms with high-yielding cows had the highest FWAS. It is certain that financial benefit consequent to good welfare is an important motivation for the farm workers. However, a Danish study showed that it is not always the main driver, and focusing on more than cost alone is important when initiating changes on the farm [44].

In general, both veterinary practitioners and farm workers demonstrated high WAS on topics related to cattle characteristics, such as social structure of adult cows, calves’ social needs, and the cows’ senses. The effect of human behavior on milk production was also well understood, mainly due to extensive training efforts regarding milking management by the Israeli Dairy Board through the years [45]. However, milk yield is affected by numerous factors—mainly genetics and nutrition—and veterinary practitioners might have a better understanding of the complex connection between milk yield and welfare. While most practitioners understood that high milk yield does not necessarily signify suitable cow welfare, only a third of the farm workers seemed to hold this notion.

Most practitioners and farm workers agreed that cows can identify specific humans connected to a positive or negative experience; 78% of the practitioners thought that cows remember previous events from an early age, and they all thought that experience with humans would affect the nature of the adult cow. Although only 60% of the farm workers agreed that cows remember past episodes, 80% thought that their interactions with humans would affect their temperament as adults.

Awareness of the importance of calves’ social rearing has progressed greatly in recent years. In our survey, 90% of practitioners thought that calves should be reared in pairs or groups until weaning, and only 4% preferred isolated rearing until weaning. The main reason for individual rearing of calves in Israel is disease prevention [2]. Farm workers seemed to be uncertain on this topic, as 40% of them thought that individual rearing is preferable, even if the calf is healthy.

Looking at the perception of cows’ pain in both groups, survey results demonstrated that farm workers tend to think that cows are less likely to mask pain, as opposed to the veterinary practitioners, who are more familiar with the instinctive pain-masking prey behavior in cattle. Nevertheless, in the case of visible pain, such as severe lameness, 80% of the farm workers understood that this would affect milk production. Thomsen et al. [46] found that even though farmers consider most types of morbidity in cattle to be slightly more painful than veterinarians, they are less likely to use analgesics [46]. Interestingly, according to our survey, farm workers are willing to use pain relief when they are aware of the cow’s suffering (such as during disbudding or cold branding). Hence, early identification of pain by the farm workers is crucial for efficient treatment and good prognosis and largely depends on proper training by the veterinary practitioner in the field. Pain relief medication is not yet compulsory in Israel, but it is being increasingly used on farms [2]. In the survey, all veterinary practitioners thought that pain relievers should be administered for disbudding, and 90% agreed that this is also essential before branding. However, only 63% stated that they had specifically instructed the farm workers to use pain relievers. Survey results are consistent with previous findings, emphasizing the practitioner’s role in setting up herd health management programs, especially in the face of upcoming challenges [8,47]. Although veterinary practitioners have a broad knowledge base, they are generally not trained in communication skills [41], making information assimilation more difficult. This gap between professional opinion and the instructions given on the farm raises ethical concerns and requires a more in-depth examination.

In summary, this study found that there is considerable variation in welfare knowledge and practical applications, and this shows there is scope to improve dairy calf and cow welfare. Furthermore, many of the aspects of welfare knowledge link to good farm and animal husbandry, and there is good evidence that welfare knowledge is associated with levels of cow productivity across farms. Therefore, it is in the interest of both the animals and their managers to improve the knowledge for application of best practices.

## 5. Conclusions

The aim of our survey was to explore the attitudes toward practical animal welfare issues on dairy farms. The high enjoyment obtained by farm workers and veterinary practitioners from their work and their perceived importance of cattle welfare constitute the elementary foundations for continual improvement. Professionally, farm workers were relatively proficient on the effect of welfare awareness on productivity, knowledge regarding cattle’s senses and social structure, and the effect of man–animal interactions on milk yield. There is room for improvement in three main areas: farm workers’ awareness of cows’ learning and memory, farm workers’ awareness of pain masking, and knowledge transfer from veterinary practitioners to farm workers. Since the veterinary practitioner is a leading professional on the farm, a great deal lies on his or her shoulders. Although farm workers give high importance to animal welfare issues, some seem to hold to traditional habits and changing their perceptions is rather challenging. The results highlight the importance of communication and understanding between farm workers and their veterinary practitioners. In light of the gaps demonstrated by the survey, reasons for insufficient knowledge transfer have yet to be understood. In conclusion, the results emphasize the variation in welfare knowledge and practical applications across farms, and it is in the interest of both the animals and their managers to improve the applied knowledge of good practices.

## Figures and Tables

**Figure 1 animals-11-00294-f001:**
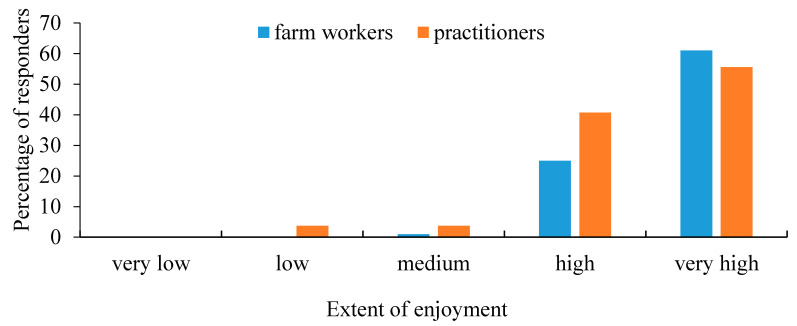
Extent of the veterinary practitioners’ (*n* = 27) and farm workers’ (*n* = 429) enjoyment from their work on the farm. There were statistical differences between groups.

**Figure 2 animals-11-00294-f002:**
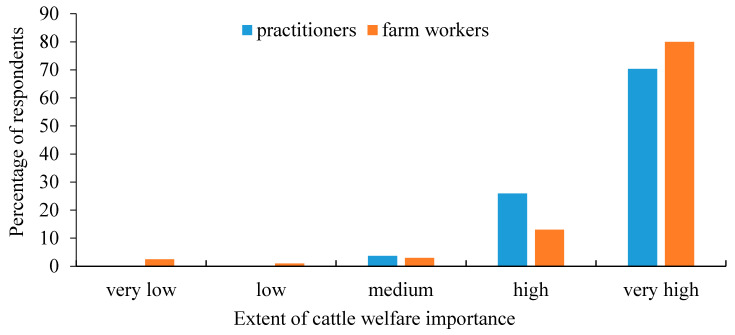
Importance of cattle welfare as perceived by veterinary practitioners (*n* = 27) and farm workers (*n* = 429). There were statistical differences between groups.

**Figure 3 animals-11-00294-f003:**
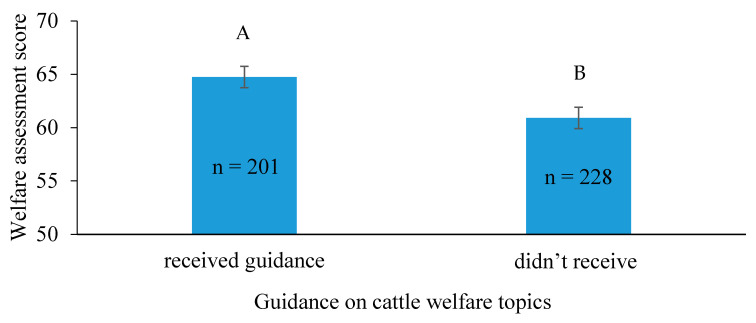
Effect of animal welfare guidance on farm workers’ (*n* = 429) knowledge (*p* = 0.02), as expressed by the welfare assessment scores, presented as least squares (LS) means ± SE. Letters “A” and “B” signify statistical differences between groups.

**Figure 4 animals-11-00294-f004:**
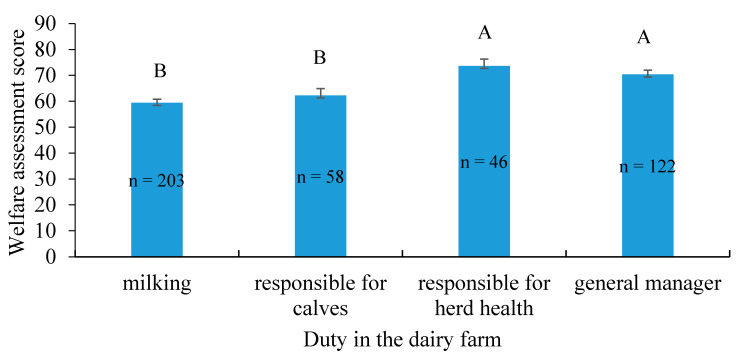
Welfare assessment score (LS mean ± SE) for individuals with different duties on the dairy farm (*n* = 429; *p* = 0.02). Letters “A” and “B” signify statistical differences between groups.

**Figure 5 animals-11-00294-f005:**
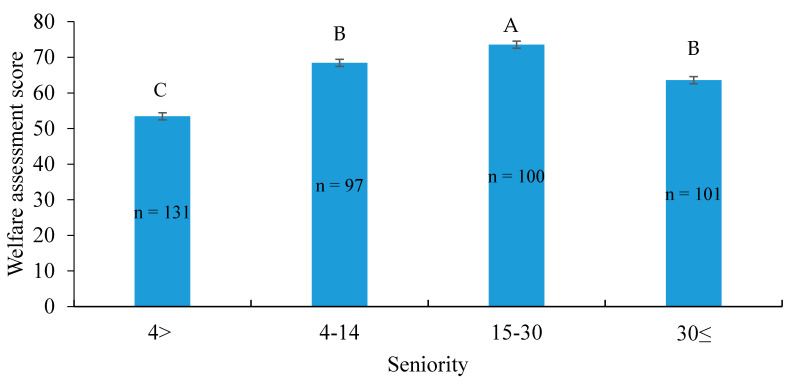
Effect of farm workers’ seniority (years of experience in farming) on their knowledge of animal welfare, as expressed by the welfare assessment score. Values are presented as LS mean ± SE (*n* = 429; *p* < 0.0001). Letters “A”, “B” and “C” signify statistical differences between groups.

**Figure 6 animals-11-00294-f006:**
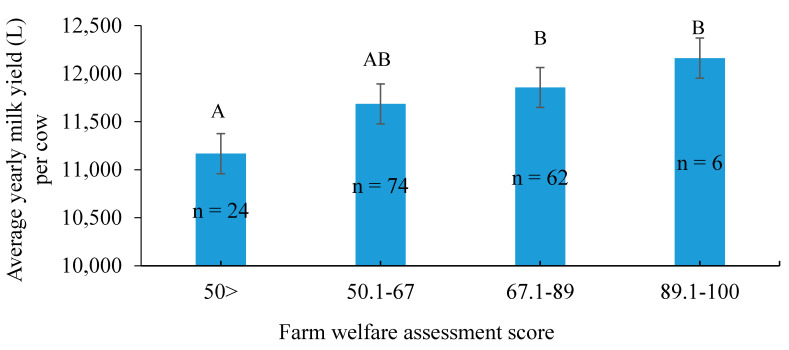
Effect of farm welfare assessment score (FWAS) on milk yield (corrected to 305 days) on Israeli dairy farms (*n* = 166). Values are presented as LS mean ± SE. Letters “A”, “B” and “AB” signify statistical differences between groups.

**Figure 7 animals-11-00294-f007:**
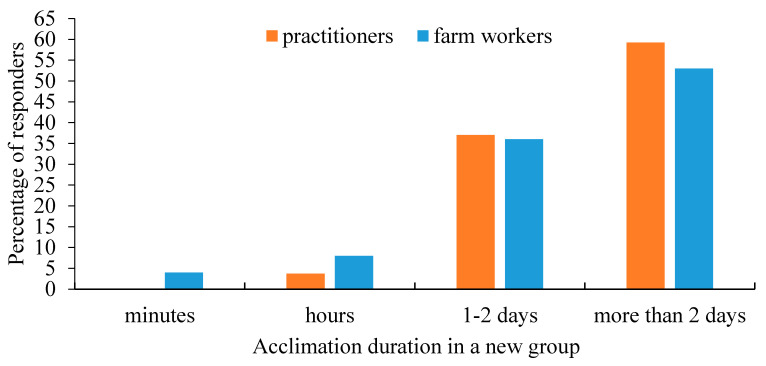
The distribution of the farm workers’ (*n* = 478) and practitioners’ (*n* = 27) assessment of required acclimation duration in a new group of unfamiliar cows. There were no statistical differences among groups.

**Figure 8 animals-11-00294-f008:**
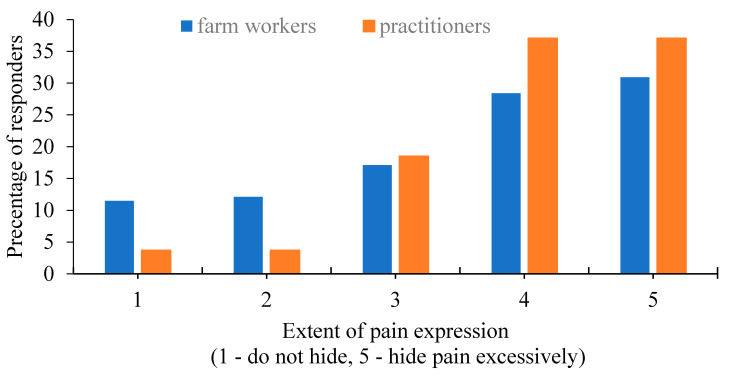
Farm workers’ (*n* = 479) and practitioners’ (*n* = 27) assessment of pain expression in cows. There were no statistical differences among groups.

**Figure 9 animals-11-00294-f009:**
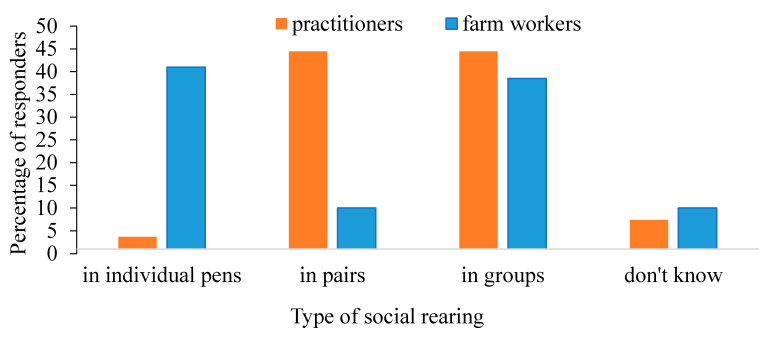
Distribution of practitioners’ (*n* = 27) and farm workers’ (*n* = 482) opinions regarding the preferred type of social rearing of pre-weaned calves (up to 2 months of age). There were no statistical differences among groups.

**Table 1 animals-11-00294-t001:** Distribution (in percentages) of veterinary practitioners’ answers to questions regarding senses, memory, and pain perception (*n* = 27).

Questions	Yes	No	Don’t Know
1. Does the cow hear things that we cannot hear?	92.6	0	7.4
2. Do cows see as we do?	3.7	92.6	3.7
3. Do cows recognize people who have treated them negatively or positively in the past?	77.8	3.7	18.5
4. Does a cow remember things that happened to it as a young heifer?	77.8	11.1	11.1
5. Does a human’s behavior toward a calf affect that calf’s behavior toward humans as an adult cow?	100	0	0
6. Will a cow that is in pain necessarily show a decrease in milk production?	3.7	96.3	0
7. Do you think that it is essential to give calves painkillers when removing their horn buds (* disbudding; with the use of caustic paste)?	100	0	0
8. Do you think it is necessary to use painkillers after cattle surgery (gastric volvulus, cesarean section, etc.)?	100	0	0
9. Do you think it is necessary to use painkillers for severe lameness?	100	0	0
10. Do you think it is necessary to use painkillers for mastitis?	100	0	0
11. Do you think painkillers are necessary during cold branding (at the age of 4 months)?	92.6	7.4	0
12. Do you instruct the dairy farmer to use painkillers when branding?	63	37	0

* Performed up to 2 weeks of age.

**Table 2 animals-11-00294-t002:** Distribution of dairy farm workers’ answers to questions regarding cow senses and memory, human–cow interactions, and pain perception (*n* = 483).

Question	Yes	No	Don’t Know
1. Does the cow hear sounds that we can’t?	334 (68.7%)	38 (7.8%)	114 (23.5%)
2. Do cows see in the same that way we do?	124 (25.7%)	303 (63%)	54 (11.2%)
3. Can cows recognize people who treated them negatively or positively in the past?	337 (69.7%)	62 (12.8%)	84 (17.4%)
4. Does a cow remember experiences it had when it was a young calf?	273 (56.8%)	96 (20%)	112 (23.3%)
5. Does a person’s behavior with calves affect the calves’ behavior as adult cows?	379 (78.8%)	48 (10%)	54 (11.2%)
6. Can the milker’s behavior “annoy” a cow during milking?	450 (94%)	23 (4.8%)	6 (1.2%)
7. Do you think it is important to give a calf pain killers when removing horn buds?	348 (71.7%)	81 (16.7%)	56 (11.5%)

**Table 3 animals-11-00294-t003:** Distribution of farm workers’ responses to assumptions regarding pain, and effect of pain on milk yield (*n* = 483).

Assumption	Agree	Don’t Agree	Not Necessarily	Don’t Know
1. High milk yield signifies good animal welfare	275 (57.3%)	13 (2.7%)	174 (36.2%)	18 (3.8%)
2. A cow experiencing pain will show a drop in milk production	295 (61.6%)	43 (9%)	121 (25.3%)	20 (4.1%)
3. Lameness affects milk yield	372 (77.5%)	22 (4.6%)	68 (14.2%)	18 (3.7%)
4. Cold branding is a painful procedure	303 (63%)	50 (10.4%)	88 (18.3%)	40 (8.3%)

## Data Availability

The data presented in this study are available on request from the corresponding author. The data are not publicly available for safeguarding individual rights of farm workers and veterinarians.

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
