# Peer review of "Welfare Issues on Israeli Dairy Farms: Attitudes and Awareness of Farm Workers and Veterinary Practitioners"

_animals, 2021, doi:10.3390/ani11020294_

Round 1

Reviewer 1 Report

This is a well written paper that does not say much that is new but what it does say is useful

Reviewer 2 Report

Overall, this work looks useful, with a large sample size for the survey amongst the farmworkers. As Israel has a high proportion of highly productive and more intensively managed dairy herds it is an interesting exemplar of the impacts of the intensification process for a wider audience. The sample of veterinarians is much smaller, but relevant and the combination of farmworker and vet provides good  opportunities for debate/discussion about their inter-relationships.

General comments

  • A minor point but farmer is used 115 times but ‘farm worker’ only once. Given the scale of dairy farming in Israel and the typically large size of herds then there are multiple farm-workers on dairy farms. As I read the paper, virtually all references to ‘farmer’ would be  better as ‘farm-workers’ (or some other  equivalent such as dairy-workers). It certainly reads as the majority of respondents are workers, employed etc. In standard UK farming circles, the ‘farmer’ would refer to the owner-occupier, head of family on family farms, main manager on larger farms, and employees across the range from milkers, calf-rearers and general farm workers would be  best be referred to as farm-workers (or similar) throughout. To provide context and descriptions of  who the respondents to the surveys were, it would be good to have a 1-2 sentences on farm and employment structure on the  dairy farms in Israel, and to provide clear indications where different cohorts within the sample of 500 fitted in. At one point (L161) mention is made of 500 farms, which implies only one respondent per farm, but eventually it is clear it is 500 respondents from 174 farms.  How multiple respondents per farm are dealt with needs to be much clearer, and the validity of these descriptions needed to be checked

Also some potential misunderstanding in nomenclature with veterinarians, alternatively referred to as ‘practitioners’ e.g L101. Suggest keeping to the single term throughout.

  • Conclusion, Discussions and some the introduction seems to stretch the evidence too far from both the literature and the study itself and take a somewhat more advisory/recommend heavy wording than can be supported by evidence. I have looked at some of the literature that I knew, and some I did not, but found generally poor referencing practice throughout with often the referenced publication not providing the evidence it was used for within the text. Any rebuttal would need a section confirming briefly where/how the reference did provide the evidence written within the paper, before I would have confidence in the referencing.
  • Much of the Introduction, and the references, are not strictly necessary, providing much about the behavioural science background of dairy cow husbandry. However, little is provided about Israeli dairy farm herd and management structure and who works on them and their training levels. Despite much of the Introduction being about the behavioural background, for example in the development of behaviour in your dairy calves, in the Results and Discussion sections whilst noting the beliefs of vets and farm-workers, there is no clarity of who is the most correct, when neither group rarely has a 100% agreed answer.
  • There are virtually no statistical analysis of the differences between the vets and farm-workers, even though there are often comparisons made.
  • There is at least one example where correlation is linked to causation
  • Rarely does the paper provide precise descriptions of the question asked of either vets or farmers, or any context. Provision of the text of the survey/questionnaire could be provided in a Supplementary section.
  • The core measurement, the ‘score’ which occasionally is given as ‘welfare score’ and once as ‘welfare awareness score’ is not adequately described in the Method section. The two sentences covering it in the Methods, I found impossible to understand. Specifically, there appears to be both a score for each respondent, but often there were much less than the 500 for no given reason, and for ‘farms’. For the latter, with multiple respondents for many farms, there appears to be no methodology explaining how a single farm score was ascribed.

Specific comments

L20 ‘The main professional figure..’  . Is there evidence that vets fill this role ? ‘A major professional….’ Or ‘the main ..on many farms..would be much easier to support. Alternative professional figures might be management consultants, nutritional consultants and even AMS technicians…

23-24 ‘Results demonstrated that..enjoyment …was…to cow…welfare’ This is a key conclusion. Need to check this is supported by evidence, if it included in abstracts. Oddly it is not found in the Conclusion.

26  ‘..requiring improvement were..’ This statement is quite strong and more appropriate for changes to regulation or guidance than a scientific conclusion. ‘…welfare might benefit….’ Or ‘…areas where welfare might improved…’ might be preferabe The key element of wording is a sentence including  ‘might be’ rather than statement such as ‘requiring’……. 

35 ‘   .awareness of animal welfare had a positive effect on milk yield..’ I do not see you have provided evidence for this. You have shown some linkage between welfare awareness and milk yield of the herd. But your statement mixes a correlation with causation a basic logic/statistical issue. Higher yielding herds may have many different characteristics compared to lower yielding including who is employed there, and their knowledge.

  1. ‘Veterinarians advocated……,but only 63%...’ This sentence needs careful consideration based on evidence from the survey… If you are going to quote the 63%, then should also note , ‘all or %’ at start of statement

Key words  ‘ you are using same words as in title. Key words should be different – check advice from journal

  1. the statement ‘One of the important changes…’ needs evidence, via a relevant reference. Actually a reduction in the number of farms per se should have no impact on the welfare of cows on the remaining farms. Changes in farm/herd size are relevant to this paper. However, rather than an unsubstantiated statement that welfare is influenced by farm size, either of both, use references from literature that supports the statement, and in absence of  hard evidence then better to say ‘A potential issue..’ or similar because at this stage in an Introduction to a paper, you do not know this

54 ‘….helps promote..’ this wording seem odd. The whole sentence should be reviewed.   Whilst both references referred to include sections on precision technology and links to welfare, it is not clear to what extent they support this sentence as it currently reads.

The case is being made here that ‘science-based knowledge’ by presumably farm-workers, is important for good practice in relation to cow welfare, and some of that is associated with precision technology and data. It just needs writing, and supporting better, by the literature

62-63  ‘….could be beneficial to farmer and cow..’  The Harris reference [7] is about social structures and herd and sub-herd in very extensively grazed beef cows and whilst the reference does state that ‘Incorporating knowledge of cattle social behavior should improve management of cattle on the range’ the link of that paper to the statement about how knowledge of cow social behaviour by farm-workers could benefit Israeli dairy herds is unclear. The de Freslon reference is about social networks and disease spread and it is not clear again how this relates to the level of knowledge re social behaviour in the survey questionnaire..

68-73 I am not convinced that these two paragraphs  has great relevance to the paper.

79 ‘Cow can differentiate dairy farmers whose behaviour is positive.. ‘  not sure how this statement is referenced, Neither [19 or 20] appears to have looked at individual person recognition by cows or grouped by general approach, though one asked the stockpeople what they believed  about cow recognition

The results of this work does not appear to support the term ‘greatly’ , some evidence  of effect, omitting ‘greatly’ is recommended..

82 The de Passille reference [20]  is about dairy calves and shows some recognition of people handling them in different degrees of positive and negative manners. This reference is linked to the statement about ‘personal feelings’. This is an unsupported term and is not used by the paper which is referenced.

87 The Rushen reference [24] is with adult cows not calves as described at line 87.

  1. The Jensen [30] reference relates only to calves up to 6 months of age, but the paper text uses it to refer to ‘throughout life’, saying ‘cow…easier to manage’ – inappropriate referencing

Lines 95-98 Three studies in NZ, Sweden and UK respectively. The UK paper does show underuse by veterinarians, not as implied in the paper on farms i.e. by farm-workers.  Reference should be unambiguously used correctly..

L101 suggest a new paragraph for this section summarising  what the reported  study did as it does not follow directly, and  does  not relate only/mainly to the topic  in the paragraph start (pain).

Missing from Introduction

  • Context about the structure of Israeli dairy farming with references.
  • Context about dairy farm labour on Israeli dairy farms, linking later to who the people were who answered the survey. Similar for veterinarians in Israeli.
  • It is evident from Methods that Arab speakers and also Thai speakers were likely contributors to survey, this may need explaining of where these group fit into farm ownership, management and labour roles for both Israeli, but definitely global readers
  • Any similar surveys with dairy stockpeople elsewhere and in Israel.
  • Survey methodologies

Methods

130 Categorization into cooperatives and  private using threshold of 300 cows. There is no justification for this threshold. It is unclear whether any questions about the farm of the participant were asked (from info provided about q’aire), and if not, where was information about farm size obtained?

L140 Previous audits and guidance from the Ministry of Agriculture are mentioned here, but not covered at all within the Introduction. If these are published, or were provided confidentially for the authors then this should be covered in the Introduction and relevant points summarised.

The questionaires themselves should be included in the paper, perhaps in a Supplementary section.

148  “A summarized score, based on the scientific literature, was given to each questionnaire to examine the effect of attitude and awareness on different parameters” This gives virtually no information about what was done to the data. Mention is made of scientific literature here but the Introduction makes no mention of  survey methods or results.

150  “In cases of personal opinion, answers were not included in the scoring”  Again, it is unclear what was done here as all information provided by respondents was their personal opinion, that is what the survey is about.

151 “The effects of the survey results on each farm and its production data (taken from the national herd database) were examined” Again this is lacking in any detail and is difficult to understand. Arguably the farm data might have been related with some of the survey results, but not the other way around as written here, with  the survey itself having an effect  upon each farm.

Missing: no information is provided on how the farms were found, and within multi-employee farms, how each(and which) individual was asked to complete the survey.  Line 161 states ‘Herd =500 dairy farms’, so this statement needs an explainer, presumably this was the data added to each person where n=500, but not for any farm-based

This section needs a complete re-write being clear what was done to the survey results in a logical order, such as confirming how, and what ‘farm’ data, was combined with the survey results, what data was excluded and why.  

166 ..’….final dairy farm score….’ implies a method that I do not think is explained

168 though  here ‘final welfare score’ is mentioned, probably for first time, having a maximum score of 100.

In this section, it appears to be a dual classification about firstly each survey participant, but then secondly about each farm/herd though it is unclear how the two elements inter-relate, especially into how respondents were chosen in each farm.

  1. Repeating the results given in Figure 1, with a relatively low sample size is probably unnecessary, and stating the actual score e.g. 3 on the 5 point scale for each point on the scale is also unnecessary.

Fig 1 and Table 1 legends both refer to an n=29, whilst text refers to 27. Please correct.

Figure 1 n=351 is quoted for farm-workers, why not 500? This difference is large and should be discussed if it refers to a key issue, whether this was a nil response or a ‘do not know’ or removed for other reasons.

L199 Here it states ‘500…from 174 farms’ which confirms the uncertainty above about numbers of employees per farm. There were clearly multiple responses for many farms. Yet the statistical analysis section refers to 500 farms. This implies that multiple answers per farm were somehow lost in analysis. This needs both clarity of method description and some discussion..

L205 the concluding sentence ‘farmers gave higher importance....than veterinarians’ referring to the data shown in Figure 1 is neither supported by any statistical evidence, or indeed by eye.  This is a wrong conclusion and appears to warrant removal, if anything the results show how similar vets and farm-workers are.

201-204. The enjoyment question and results in text here does not need all the scores and choices as mentioned above (e.g. the ‘3/5’). A combined Figure for Veterinarians and farm-workers would show profiles and also provide any indication of similarity/differences between farm workers and vets. For both ‘differences/no differences’ could be clearly covered.

213-214  n and a p value is given, but no actual means 

219 the term ‘lower knowledge’ is used here. It is unclear what this means, and how it is identified. If it refers to previously mentioned ‘scores’ this should be explained in the Methods.

219 The two classes here, ‘on-farm’ versus ‘courses and workshops’ should make it clear where those have both categories fitted in and the term ‘only might help. Reference back to the questionnaire would help to clarify what these terms mean.

222 again clarity on whether these questions could be mutually exclusive (i.e. one only) or many/all of them and how the results were classed is missing. It is likely many farm-workers get multiple sources of guidance on animal husbandry.

Figure 2 The formatting will likely need changing to a simpler line/box graphic. The statistical difference notations (A, A, B) is likely to need formatting according to Publication guidance but should likely be foot-noted in all tables/figures where used.

N=350 here, which is much different from the sample of 500 and it is not clear why.

231 please write more correctly e.g. …among those with different farming duties.

233 ‘suckling’ calves usually refers to calves still suckling upon their dam, but here it is unclear what is meant here, possibly ‘young calves’ is sufficient as it might mean the calves being artificially reared on milk substitutes etc..

Figure 3 refers to ‘welfare score’. This is not mentioned clearly how it calculated with mention of ‘score’ at line 149 for example being under-described. No discussion of the term ‘welfare score’ appears to be made. Other figures have n values and figures/tables, but not here and all should be consistent. Cf earlier note re notation on statistical differences.

246 The title of this section should be changed, as is confuses possible correlation with causation and something like ‘Links to farm productivity’ would be better

248 method to calculate ‘farm score’ should be given in Methods and then clearly labelled throughout. Use of the term ‘score’ without any classification should be avoided as there are a number of different scores appearing.  It remains unclear throughout how score for farm workers differs from score for farms in terms of how these are calculated when there are single and multiple respondents per many farms.

248-255 The way these results are portrayed, splitting arbitrarily into different classes (e.g. 67.1-89) is odd. Data could be alternatively be considered by a variety of regression techniques of the individual scores and individual farm milk production data and farm size etc. The later should be clearly/correctly labelled in axis title/legend and maybe in footnote – it needs to refer to per cow per year and mention of 305 day corrections

257 It would be better to summarise the question itself rather than the category. For the farmer for example it is unclear whether they are being asked for answers about their beliefs or what they actually do in terms of arm practices

Figure 6 Again, there are issues with n with now 478 farmers and 29 vets, why not 27? Without opportunity to look at the question itself it is difficult to be clear what was asked, and thus what the figure actually means.

265 Providing a clear description of the context of both question and answers in this way is better than used earlier. However, the question itself is rather odd in terms of behavioural science meaning. There appears to be a set of optional answers which are confusing  themselves if a large number of respondents answered ‘no hierarchy in cows’.

267-274 Again this section raises more questions than answers about wording of  question itself, and the choices available to respondents. The full questionnaire including questions and whether  single only or multiple choice options were given needs to be provided and the most important  issues summarised in the Methods section.

294 table 2 is a good table, but then some new data is added in the text here. Suggest it all goes into the table

314 this sentence is referring to 2 different types of answer, about painfulness and about requirement  for pain medication. Ideally, the answers by both groups to both  questions would be helpful. Either way separate the issues clearly as they are not the same thing

320 this sentence covers  farms, earlier we are told n=174, and  this data e.g. 14% with a n=23 gives a total of 164 farms. If there are multiple employees for many of the farms, it is not clear how any conflicting answers were resolved to provide 1 answer per farm. Clarity in Methods is needed.

322 Suggest replacing ‘agreed’ with ‘believed that’ . This sentence  mixes  up age with stage in production system because the inferred question is about age, but the answer is about weaning and whilst the two may be related and quite standard in Israel, this tells us little about the precise questions and the precise answers.

325 ‘Conservatively’ – this word looks the wrong English.

Figure 8 shows quite large differences between farm-workers and vets, so some statistical analysis looks warranted. A Chi2 test or other  contingency table method would  be useful to do, and confirm result.

333 ‘full’ is odd language here suggest delete

333-337 Clearer wording of the questions and clear wording of the answers are needed here (And elsewhere). It  is for example uncertain if the full question is about avoiding painful procedures in the milking parlour and it is unclear whether the farmers were specifically not following specific instructions given by vet. Similar is with the use of disbudding paste.

338 this question about resolving welfare issues is important, but without clarity of the question itself, and the choice, if any, of optional answers it is rather poor methodology. Given the farm-workers involved are rather vaguely combined throughout, it would  be useful to identify to whom the answers refer when ‘farmer’ here is mentioned, is it any farm-worker or the lead farm manager/herd owner, for example.

345 This paragraph if it part of the motivation for the study might best be at the end of the Introduction

351  Insert ‘many’ as unless there is evidence otherwise, then this is not true. 24hours monitoring is certainly not the case for many dairy systems. This paragraph if needed, should be in the Introduction, it does not fit here.

356 and 357 ‘welfare scores’ and ‘welfare awareness scores’ are used here. The latter term appears  here for  the first time and it is not clear whether these are the same thing and as noted earlier it is not well-described how this is calculated. Elsewhere ‘farm scores’ and just ‘scores’ are mentioned. Also all unclear whether the same of different.

‘seniority’ is also a little unclear as used here it is not clear whether this is age, experience or about hierarchy/responsibility in those farms with a management structure.

358 this statement is made  speculatively, but the study appears to have information about the training profile of individual respondents (and the survey is about individual respondents not about farms)  from the different types of farms, and certainly could class farms by herd size. After noting there is a difference, then the speculation as to why becomes possible.

361…the scoring gap is not detailed in  the text /tables or I could not find it . referred to at 213-214 but no means scores given. The difference by statistical test is marginal. The different structures of the farms, and who answered the survey may be relevant and there is no data presented showing how the profiles of respondents differed by farm size.

362 the solution proffered does not  seem likely to solve the reason speculated, i.e. lack of time, and is not really useful at this point. A single paragraph about actions for whole paper may be useful, but section by section seems not appropriate for a scientific paper .

365 the missing part of the argument is clarity on whether the authors believe the higher scores (whether these are farmer scores or farm scores) are about welfare awareness or also transfer into farm action and thus improved animal welfare in practice. Does the paper provide any evidence of this? This seems an important question linking the scores to the actions and then after that to potential higher productivity of the herd.

373 there are no references to the Israeli Dairy Board, and no indications of any evidence of their involvement so the link of them to this specific section is unclear. Could be included in Introduction

373-375 This sentence, and what the second part is trying to say is very unclear. Within it ‘practitioners tend..’ appears to be an un-evidenced speculation about something. Is it vets compared to farm-workers, and who is defining ‘better understanding’ as a higher score is mainly about beliefs, not knowledge of animal husbandry.

375-377 This sentence makes another assumptive comparison between vets and farmworkers, without clarity on what it means. The results show that farmers and vets have different views/beliefs on links between milk yield and welfare – its quite likely they think words like ‘welfare’ mean different things (and there is literature that evidences issues like this). For these comparisons, in which a difference between groups are stated, or implied, there is no statistical evidence offered to support, and this needs some narrative of stats, or failure to find differences or some other justification for not providing statistical support for the statement given.

386 ‘The main reason…’ It is not clear whether this statement refers to answers from the survey, and if so by vets/farm-workers or both, or is just a general comment.

387 The term ‘conservative’ does not tell the reader much here. There looks to be a difference that would be shown up by a probability analysis such as Chi2 . The ‘…even if the calf is healthy’ statement, suggests that a scenario was given in the survey relating to health of the calves. If a scenario such as ‘healthy calves…’ was not given in the survey then this sentence section should be removed.

389 This sentence is about advice and solutions to problems and not a scientific conclusion

402 This section implies that whilst 90% plus of vets thought some operations were painful, only 63% stated they had  specifically instructed pain relief is complex and  not helped by lack of availability of the background paperwork of the precise questions.  There is a large difference between 63% always, sometimes or occasionally and suggesting, advising and instructing.

413 The logic that because farmers enjoy their work, that they are open to change, new knowledge etc is not clear. There are clearer, and much greater evidence for  the willingness, indeed drive, to change and adapt on dairy farms though adoptions of new technologies and practices for e.g.

415 is there evidence that vets are ‘the’ leading professionals on dairy farms? Elsewhere is world, other consultants sometimes have greater influence and without evidence, then replacing ‘the’ with  ‘a’ is suggested.

415-416 and then 1 line on from 413, the ‘open to change’ argument is reversed stating farmers are conservative and less likely to change. Indeed as this survey did not appear to cover change in factors specifically related to animal welfare, then this point is not evidenced through the results.

Conclusions might be better focussed upon confirming importance of welfare by both parties and that very similar. Anand where agreement and differences appeared between farm-workers and vets and what should be done if differences are real and animal welfare would improve if something could be done about farm-working training and awareness, and about awareness by vets of farmer actions (do they know that many farmers do not use pain relief?).

465 Dawkins M.S. is the way Marion Stamp Dawkins is usually referenced including in her own papers.

Apologies if I have mis-read sections and misunderstood, but as ever, this is also the result of poor explanations.

Reviewer 3 Report

This is a really interesting paper with some very useful conclusions that clearly outlines gaps that need addressing. The introduction provides a comprehensive review. However I do take issue with line 77-78: productivity may be commonly used as an indicator for welfare assessment but it is seriously flawed - intensively reared battery chickens may produce lots of eggs but do not have good welfare - so this sentence needs some expansion. Lines 148-152 would benefit from further explanation to make the methodology repeatable. 

Round 2

Reviewer 2 Report

Thankyou for your response. I am pleased how you have dealt with my comments and the very significant changes throughout remove most of my reservations about the paper. The methodology is now much  clearer and simple things like including the questionaires directly are very helpful to understanding what was done both in the survey, but also in terms of analysis and presentation of results. Overall, I now consider the paper worthy of publication.

I have mainly minor comments by line numbers, though there are a couple of more significant issues

L130 please state somewhere early in Methods for both vet and farm worker surveys that the survey is provided in Supplementary

L158-161 I understand the split between the farm types; coop and family-farm. I am just not wholly clear on how you describe it here. So <300 cows are classed as private family farms. I doubt very much that the type split is exactly at 300 cows, so clearer language (and ideally a short sentence of explanation) about the split should be used. It is written here as if all farms over 300 cows were coops and all farms under 300 were family farms. Unless they were selected to be so then clarity is needed. And if selected on basis that had to be right for both cow herd number and family/coop type, then this should be stated.  If not ‘..the vast majority of these farms were cooperatives…though with x% of these being family-owned. Of the <300 cows herds x % were cooperatives’ or similar. I presume this data exists to accurately class whether family or coop without the cow number boundary. It might be that <300 and >300 might be a more accurate title header.

Line 159. As per previous review you have not stated how the 174 farms were selected, was it random or some other method? This sentence was difficult for me to understand initially…so I would suggest a different running order for Methods sections such as  “Sixty-one and 113 coop ……family farms were selected from all geographic regions, representing 37 and 20% of each category, respectively, in the national ‘x’. or similar. Where ‘x’ says where the full dataset comes from, presumably it is a national database where the list of farms were registered. It would be interesting to note whether the survey was voluntary or obligatory and whether there was any resistance or failure to comply.

L178-182. I think this new section that explains how each farm worker WAS is made could be looked at again.  I would order it as ..18 ‘yes/no’ questions….; Q5 and Q13 excluded; ‘correct’ and incorrect;  ‘correct’ given 5.55 and ‘incorrect’ given zero; total WAS score out of 100.

L224 etc . The order that Figures 1 and 2 and Table 1 are given are not in the order given in the text and their position in the paper. The journal will probably sort this out, but better to construct the paper in the correct order. Figures 1&2 together makes good sense, even alongside.

L242, 243, 244 . there are a succession of ‘most of..’ statements followed by a statement that is less than 50%. So ‘most of workers (39%)’ etc. This statement is not correct in English usage, and should be more precisely labelled as ‘the most frequent category was ..<30 years, i.e. young workers’ or similar.

L305 title effect upon milk yield

This issue arose in the earlier review. In version 1 I criticised draft that it was making a causal link between the WAS/FWAS scores (As re-named and that works very well) and herd milk yield and making a causal link rather than noting a correlation. I believe this heading title should be changed. Something like ‘Relationships between…’ or ‘Links between..’ would be supported by the evidence you provide

As noted before you have no evidence that higher welfare scores have an effect upon herd milks yields. But you do have evidence that the two factors are related or linked. The references in the Introduction now point to studies at herd and individual cow scales where direct impact of human behaviour and milk yields are related.

L370 Tables 1 and 3 and Figure 8. I note earlier in response to V1 that authors state that a statistical test comparison was not warranted because of the large difference in sample sizes n =27 and 479. The abstract effectively, at L43, states that the visual differences seen are real, so some statistics would ideally be done to show this. I did a quick contingency table test, grouping 1-2,3,4-5 and estimating values from Figures. See below. These results suggest that the visual differences are not supported statistically. A Fisher Exact probability test (good for low n numbers) for 2 x 2 (omitting ‘mid’) is also non-significant, also suggesting no statistical significance with the low number of veterinary practitioners. I appreciate you are working with the data you have, but really need to be careful of stating differences that do not have evidence.

Figure 8 legend  n=29 – this should be n=27?

L389 Sorry but your changed sentence start ‘By virtue of habit..’ is not particularly good English and unclear to me, I presume it is about traditional practice. It is the Results section, so best not to speculate without evidence from data, so maybe just stating the result would be best.

L440-447 Despite what I have noted about the title at L305, I think the narrative and conclusions about relationships between productivity, whilst not providing causal evidence of the link between welfare knowledge and herd average milk yields, could be strengthened. This paragraph here is quite short and particularly there is no ‘link between welfare and productivity’ in conclusions of abstract. The results summarised in Figure 8 and added to elsewhere do indeed, as written by you, ‘bolster the argument’ that good welfare knowledge, and application, ‘does contribute to a farm’s success’.

It seems to me that there is scope for a final sentence or two in the discussion, also leading to a shorter Conclusion sentence and Abstract sentence or phrase that could note something along the lines of

‘This study has found that there is considerable variation in welfare knowledge and practical applications, and this shows there is scope to improve dairy calf and cow welfare. Furthermore, many of the aspects of welfare knowledge link to good farm and animal husbandry and there is good evidence that welfare knowledge is associated with levels of cow productivity across farms. So, it is in interests both of the animals and their managers to improve knowledge and application of best practice.

L484 referencing style for the two sets given here in text. Nothing wrong with  using the names of authors in the text, even if the Animals preferred format is [nn], but in this case needs dates and would suggest to [47] goes after Svenson et al, 2018 [47] but please check journal formatting rules

Figures - not some have different A, B, AB for whether statistically different, others do not and none note that pairs of columns with  different letters are statistically different. Should check journal guidelines and be consistent for all.

Whilst checking out some of your references, I noted the careful way other papers wrote their conclusions about links between dairy farm welfare issues and productivity

“ as in pigs”….“….sequential relationships may exist between the attitude and behaviour of the stockperson and the behaviour and productivity of commercial dairy cows” Breuer et al 2000

So correlations but careful phrasing on direct links

“These results indicate that cognitive-behavioral interventions that successfully target the key attitudes and behavior of stockpeople that regulate the cow's fear of humans offer the industry good opportunities to improve the productivity of cows.

From Hemsworth et al 2002 – again care in conclusions being drawn from results with some links/relationships between changing behaviour of stockpeople and resulting milk production..
